# A Short 5′triphosphate RNA nCoV-L Induces a Broad-Spectrum Antiviral Response by Activating RIG-I

**DOI:** 10.3390/v14112451

**Published:** 2022-11-04

**Authors:** Ziyang Song, Qian Wang, Lianlian Bian, Chaoqiang An, Bopei Cui, Qunying Mao, Xing Wu, Qian He, Yu Bai, Jianyang Liu, Lifang Song, Dong Liu, Jialu Zhang, Fan Gao, Xiuling Li, Zhenglun Liang

**Affiliations:** 1Shanghai Institute of Biological Products Co., Ltd., Shanghai 201403, China; 2Division of Hepatitis and Enterovirus Vaccines, NHC Key Laboratory of Research on Quality and Standardization of Biotech Products, NMPA Key Laboratory for Quality Research and Evaluation of Biological Products, Institute of Biological Products, National Institutes for Food and Drug Control, Beijing 102600, China; 3Beijing Minhai Biotechnology Co., Ltd., Beijing 102629, China

**Keywords:** RIG-I, antiviral, RNA agonist, SARS-CoV-2

## Abstract

Small molecular nucleic acid drugs produce antiviral effects by activating pattern recognition receptors (PRRs). In this study, a small molecular nucleotide containing 5′triphosphoric acid (5′PPP) and possessing a double-stranded structure was designed and named nCoV-L. nCoV-L was found to specifically activate RIG-I, induce interferon responses, and inhibit duplication of four RNA viruses (Human enterovirus 71, Human poliovirus 1, Human coxsackievirus B5 and Influenza A virus) in cells. In vivo, nCoV-L quickly induced interferon responses and protected BALB/c suckling mice from a lethal dose of the enterovirus 71. Additionally, prophylactic administration of nCoV-L was found to reduce mouse death and relieve morbidity symptoms in a K18-hACE2 mouse lethal model of SARS-CoV-2. In summary, these findings indicate that nCoV-L activates RIG-I and quickly induces effective antiviral signals. Thus, it has potential as a broad-spectrum antiviral drug.

## 1. Introduction

Innate immunity is the first line to defend against viral infection. Innate immunity recognises pathogen-associated molecular patterns (PAMPs) through pattern recognition receptors (PRRs) to induce immune responses [1,2]. In mammals, the viral RNA recognition function of innate immunity begins with RIG-I-like receptors (RLRs), Toll-like receptors (TLRs), and NOD-like receptors (NLRs) [3,4]. Among them, RIG-I is a PRR in the RLRs family. Located in the cytoplasm, RIG-I recognises short double-stranded RNA with 5′PPP [5,6,7]. Conformational change of RIG-I is induced after recognising the viral double-stranded RNA [8]. Activated RIG-I interacts with mitochondrial antiviral signal protein (MAVS) to form a protein complex [9,10]. The MAVS complex then recruits serine protein kinase (IKKε/TBK1) to induce phosphorylation of interferon regulatory factor 3 and 7 (p-IRF 3 and p-IRF 7) and activate nuclear factor kappa-B (NF-κB). p-IRF3 and p-IRF7 are transferred to the cell nucleus and induce the expression of type I interferon (IFN-I) [11,12]. IFN-I induces expression of interferon-stimulated genes (ISGs) through the Janus kinase-signal transducer and activator of transcription (JAK-STAT) pathway, directly inhibiting transcription of viral genes, degrading viral RNA, and inhibiting translation and modification of viral proteins [13,14,15]. NF-κB could mediate the expression of several proinflammatory cytokines, such as TNF-α and IL-6, which contributes to the antiviral effects [16,17].

Compared to antiviral therapy with recombinant IFN-I injection, RIG-I agonists can induce robust endogenous IFN-I responses and multiple inflammatory cytokines, maximising the likelihood of functional downstream responses [18]. Moreover, recombinant interferon, which has a different molecular structure to human endogenous interferon, may induce interferon antibodies in vivo, thus influencing the treatment outcomes [19,20,21]. Chiang, et al. [22] designed a 99 nt fragment of 5′ppp RNA named M8, which activated RIG-I and significantly inhibited multiple influenza viruses and chikungunya virus. Therefore, RIG-I agonists have significant research, development, and application prospects.

The RIG-I receptor can recognise double-stranded RNA sequences with 5′PPP [23,24,25]. Thus, in this study, a sequence with a stem-loop structure in the 5′ untranslated regions (UTR) of SARS-CoV-2 was selected and reconstructed to obtain a 100 nt fragment with 5′PPP. A hairpin was confirmed, owing to the reverse complementary pairing intramolecularly, and this small molecular nucleotide was named nCoV-L. Subsequently, the protective effects of nCoV-L against infection with several RNA viruses, including Human enterovirus 71 C2-2 strain (EV-71/C2-2), Human coxsackievirus B5 JS417 strain (CV-B5/JS417), Human poliovirus 1 Sabin strain (PV-1/Sabin), Influenza A virus H9N2 (IAV/H9N2) and SARS-CoV-2 Delta strain (SARS-CoV-2/Delta), were analysed through in vitro and in vivo experiments.

## 2. Materials and Methods

### 2.1. Biosafety and Ethics Approval

Experiments with live SARS-CoV-2 were performed in a biosafety level 3 (ABSL3) facility at the Wuhan Institute of Biological Products Co., Ltd. (Wuhan, China). All research staff were trained and qualified in the experimental procedures. Animal procedures were approved by the Ethics Committee on Laboratory Animals of the Wuhan Institute of Biological Products Co., Ltd. The number of ethical permissions for this experiment was WIBP-AII442021005.

Other experiments with live viruses were performed in a biosafety level 2 (ABSL2) facility at the National Institutes for Food and Drug Control (Beijing, China). Animal research protocols were approved by the Institutional Animal Care and Use Committee at the National Institutes for Food and Drug Control, China. The ethical permission number for this experiment was 2022-B020.

### 2.2. Cell Culture

A549 (CRM-CCL-185), RD (CCL-81), HEp-2 (CCL-23), Vero (CCL-81), MDCK (CCL-34) and C2C12 (CRL-1772) cells were purchased from ATCC. The HEK293 cell lines with *ISG54*-luciferase reporter gene (hkl-null) were purchased from InvivoGene (San Diego, CA, USA). *TLR3, TLR7, TLR8, MDA5* and *RIG-I* in the HEK293 cell lines with *ISG54*-luciferase reporter gene were knocked out with the CRISPR/Cas9 system. The backbone plasmids lenti-sgRNA and lenti-cas9-zeocin were obtained from Genescript Co., Ltd. (Nanjing, China). sgRNAs were designed and subcloned into a Cas9 backbone. All the sgRNAs used are described in Table 1. *ISG54*-luciferase reporter HEK293 cells were first infected with lenti-cas9-zeocin and then selected using zeocin. The stable sub-lines were then infected with lenti-sgRNA to specifically knock out the target genes. The CRISPR knockout cell lines were sanger sequenced to confirm frame shift (Table A1).

C2C12 cells transfected by lentivirus expressing *CXADR* gene (c2c12-CAR) were cultivated in DMEM (Gibco, Carlsbad, CA, USA), supplemented with 10% fetal bovine serum (FBS) (Gibco, Carlsbad, CA, USA) and 100 U/mL penicillin-streptomycin (Gibco, Carlsbad, Carlsbad, CA, USA) in a humidified incubator in the presence of 5% CO_2_. RD, HEp-2, MDCK and Vero cells were cultivated in MEM (Gibco, Carlsbad, CA, USA). The wild type (WT), RIG-I ^−/−^, MDA5 ^−/−^, TLR3 ^−/−^, TLR7 ^−/−^, and TLR8 ^−/−^ HEK293 cell lines with *ISG54*-luciferase reporter gene and A549 cells were cultivated in DMEM (Gibco, Carlsbad, CA, USA).

### 2.3. Viruses

EV-71/C2-2, CV-B5/JS417 (GenBank accession no. KY303900) and IAV/H9N2 (GenBank accession no. FJ499463-FJ499470) were preserved in the Hepatitis and Enterovirus Laboratory. PV-1/Sabin (GenBank accession no. V01150.1) was donated by the Influenza and Respiratory Virus Laboratory of NIFDC. SARS-CoV-2/Delta (GenBank accession no. OK091006.1) was preserved in a biosafety level 3 (ABSL3) facility at the Wuhan Institute of Biological Products Co., Ltd.

EV-71/C2-2 was used to infect RD cells at a multiplicity of infection (MOI) of 0.1. The virus was allowed to adsorb for 1 h at 35 °C in serum-free MEM. Serum-free MEM was used to wash the monolayer and then replaced with 2% FBS MEM. After seven days of infection, the medium was harvested and cleared by centrifugation (12,000 rpm for 20 min) at 4 °C. Viral titers were determined using cytopathic effect-based end-point titrations.

CV-B5/JS417 and SARS-CoV-2/Delta were propagated in Vero cells. PV-1/Sabin was propagated in HEp-2 cells. IAV/H9N2 were propagated in MDCK cells.

### 2.4. Mice

Seven-day-old BALB/c mice were purchased from the Laboratory Animal Resource Centre, National Institute for Food and Drug Control, and were housed in a specific pathogen-free (SPF) animal facility.

Six-week-old SPF K18-hACE2 transgenic mice were obtained from Beijing Vital River Laboratory Animal Technology Co., Ltd. (Beijing, China).

### 2.5. In Vitro Transcription and Gel Analysis

The sequence of nCoV-L was as follows:5′-GGUUUAAUACCUAAAAAAAAAAAAAAAAAAAAAAAAAAAAAAAAAAAAUCCCUUUUUUUUUUUUUUUUUUUUUUUUUUUUUUUUUUUUAGGUAUUAAACC-3′ (100 nt).


The target sequence was constructed in the pUC19 plasmid vector and amplified. The amplified plasmid was digested with the endonuclease *BsmBI-v2* (New England Biolabs, Ipswich, MA, USA). Then, the target sequence was collected and purified. nCoV-L and M8 were synthesised with a T7 RiboMaxTM Express Large Scale RNA Production System (Promega, Madison, WI, USA) for 3 h. RNA transcripts were DNase digested for 20 min at 37 °C and then purified according to the manufacturer’s instructions. RNA transcripts were analysed on 2% agarose gel with NorthernMax GlyGel Prep/Running buffer (Invitrogen, Carlsbad, CA, USA) for 1 h. The secondary structure of nCoV-L was predicted using the RNAfold Webserver (http://rna.tbi.univie.ac.at/cgibin/RNAWebSuite/RNAfold.cgi) (accessed on 7 June 2021).

### 2.6. Transfections

Lipofectamine 3000 (Invitrogen, Carlsbad, CA, USA) was used for all in vitro and in vivo transfections. nCoV-L and Lipofectamine 3000 reagent (1:1.5 ratio) were mixed in Opti-MEM (Gibco, Carlsbad, CA, USA) and incubated for 10 min. The volume of the mixture for in vitro transfection was 50 μL, and for in vivo transfection was 100 μL.

### 2.7. Luciferase Assays

For the luciferase assays, the *TLR3 ^−/−^, TLR7 ^−/−^, TLR8 ^−/−^, MDA5 ^−/−^, RIG-I ^−/−^,* and WT HEK293 cell lines with *ISG54*-luciferase reporter gene were transfected with 50 ng nCoV-L. The activity of the reporter gene was measured by a Dual-luciferase Reporter Assay (Promega, Madison, WI, USA), according to the manufacturer’s instructions. Relative luciferase activity was measured.

### 2.8. Protein Extraction and Western Blot

5′ppp RNA-transfected cells were collected by centrifugation at 12,000 rpm for 1 min. The cell precipitation was lysed in RIPA buffer (Sigma-Aldrich, Milwaukee, WI, USA) and cleared by centrifugation at 12,000 rpm for 15 min at 4 °C. The cleared lysis was subjected to SDS-PAGE. Proteins were electrophoretically transferred to nitrocellulose membranes and then subjected to immunoblotting analysis using the indicated antibodies. Anti-RIG-I, anti-MDA5, anti-IRF3, anti-pIRF3 Ser 396, anti-ISG56 and anti-β-actin antibodies were purchased from Cell Signaling Technology (Danvers, MA, USA). Horseradish peroxidase (HRP)-conjugated goat anti-rabbit antibodies were purchased from Cell Signaling Technology (Danvers, MA, USA). Images were visualised by using an Amersham Imager 680.

### 2.9. RNA Extraction

Total RNA in cells or tissues was isolated using a Kingfisher Flex Automatic Nucleic Acid Extractor (Thermo Fisher Scientific, Waltham Mass, MA, USA) with Pre-packaged Nucleic Acid Extraction and Purification Kits (Thermo Fisher Scientific, Waltham, MA, USA), according to the manufacturer’s instructions.

### 2.10. Real-Time Quantitative Polymerase Chain Reaction (RT-qPCR)

Extracted RNA was subjected to RT-qPCR with a PrimeScript One Step RT-PCR Kit (Takara, Kyoto, JPN), according to the manufacturer’s instructions. PCR primers were designed with PrimerBank and purchased from Sangon Biotech Company (Shanghai, China). All the primers used are described in Table 2. RT-qPCR was performed on an ABI7500 (Applied Biosystems, Arlington, VA, USA). All data are presented as relative quantification with *gadph* as the internal control.

### 2.11. In Vitro Virus Infection

For in vitro virus infection experiments, 2 × 10^5^ cells were infected with virus strains in a small volume of serum-free medium for 1 h at 37 °C. The medium was replaced with a complete medium 24 h prior to analysis.

### 2.12. In Vivo Virus Infection

Seven-day-old BALB/c mice were challenged intraperitoneally with 1.26 × 10^6^ TCID_50_ EV71/C2-2 in 100 μL of MEM.

Six-week-old K18-hACE2 transgenic mice were challenged intranasally with 200 TCID_50_ SARS-CoV-2 Delta in 50 μL of DMEM.

### 2.13. Histopathological Examination

The harvested tissues were fixed in 4% paraformaldehyde, embedded in paraffin, sectioned, and stained with haematoxylin and eosin (H&E) for histopathological analysis. An experienced and qualified pathologist confirmed the results.

### 2.14. ELISA

The concentration of IFN-β in mouse serum was determined using ELISA with Legend Max Mouse IFN-β ELISA Kits (BioLegend, San Diego, CA, USA), according to the manufacturer’s instructions.

### 2.15. Statistical Analyses

Data are presented as the mean ± standard derivation (SD). Statistical analysis was performed using GraphPad Prism software 8.0. Unpaired, two-tailed Student’s *t*-tests were performed to determine the statistical significance of differences. Differences were considered statistically significant when *p* was less than 0.05.

## 3. Results

### 3.1. nCoV-L Induce Interferon Response via the RIG-I Receptor

The predicted structure of nCoV-L is shown in Figure A1A in Appendix A, demonstrating that double-stranded stem-loop structures can be formed through complementary pairing. The 5′ppp RNA products were acquired through in vitro transcription and purification. A single stripe was identified through RNA electrophoresis detection (Figure A1B). The *TLR3 ^−/−^, TLR7 ^−/−^, TLR8 ^−/−^, MDA5 ^−/−^, RIG-I ^−/−^*, and WT HEK293 cell lines with *ISG54*-luciferase reporter gene were transfected with 50 ng nCoV-L. The expression levels of reporter gene were analysed 24 h after different treatments (Figure 1A and Figure A2A). nCoV-L activated the expression of the reporter gene in *TLR3 ^−/−^, TLR7 ^−/−^, TLR8 ^−/−^*, *MDA5 ^−/−^*, and WT cell lines, but not in *RIG-I ^−/−^* cells, consistent with the induction of a downstream factor ISG56 (Figure A2C,D), suggesting that nCoV-L can specifically activate the RIG-I receptor. When using poly(I:C) as control, no induction of reporter gene or ISG56 was detected (Figure A2A,C,D), which may be due to the insensitive of knockout cell lines to poly(I:C) compared to nCoV-L.

Western blot analysis was conducted to detect the expression and activation of proteins involved in the RIG-I signalling pathway. Increased phosphorylation levels of IRF3 (Ser396) and upregulated ISG56 were detected after transfection of 1 ng/mL nCoV-L or M8 in A549 cells (Figure 1B and Figure A2B), indicating the activation of RIG-I signalling. RIG-I, MDA5, and IRF3 were also upregulated (Figure 1B and Figure A2B), which was dependent on the positive feedback regulation [26,27]. The *IFN-β* expression levels at different times post-transfection in A549 cells revealed that *IFN-β* could be effectively induced by nCoV-L, which was higher than that of M8 (*p* < 0.05) under our laboratory condition (Figure 1C). These findings indicated that the double-stranded molecular nCoV-L, with 5′ PPP and a stem-loop structure, induced the expression of interferon and downstream ISGs to activate innate antiviral immunity by specifically activating the RIG-I receptor and mediating IRF3 phosphorylation.

### 3.2. In Vitro Antiviral Effects of nCoV-L against RNA Viruses EV-71, CV-B5, PV-1 and IAV

Next, the in vitro inhibition effects of nCoV-L against four RNA viruses (EV-71/C2-2, CV-B5/JS417, PV-1/Sabin, and IAV/H9N2) were analysed. As shown in Figure 2A, the cells were collected after treatment, and RNA was extracted to measure the viral load. The virus titers in the culture supernatants were also evaluated. The results demonstrated that, compared with the liposome negative control, 30 ng/mL nCoV-L inhibited duplication of EV-71/C2-2 in RD cells, decreasing the viral load by approximately 7.8-fold (*p* < 0.0001). Accordingly, the viral titer in the culture supernatant was decreased 22-fold (*p* < 0.0001). Under our experimental conditions, nCoV-L decreased the viral load in cells and the virus titer in the supernatants by 3.5-fold and 3.6-fold compared to the equivalent dosage of M8 (*p* < 0.01). The intracellular viral loads and virus titers of CV-B5/JS417 and PV-1/Sabin in the supernatants were assessed using the same method. nCoV-L significantly inhibited the replication of CV-B5/JS417 and PV-1/Sabin (*p* < 0.001), markedly better than M8 at the same dosage (*p* < 0.05). As shown in Figure A3, 1 ng/mL nCoV-L was also proven to inhibit intracellular IAV/H9N2 virus infection and was more effective than the same dose of M8 (*p* < 0.001). In summary, nCoV-L significantly inhibited the EV-71/C2-2, CV-B5/JS417, PV-1/Sabin, and IAV/H9N2 viral strains.

### 3.3. nCoV-L Protected Suckling Mice from Lethal EV71 Virus Infection

The stable lethal dosage of EV-71/C2-2 was determined to be 1.26 × 10^6^ TCID_50_ in a seven-day BALB/c suckling mouse model (Figure A4A) so that the in vivo antiviral activity of nCoV-L could be quantitatively assessed. As the experimental procedure shows in Figure 3A, the lethal dosage of EV-71/C2-2 was injected intraperitoneally. One hour later, 5, 10, or 15 μg nCoV-L was intraperitoneally injected. Five days post-infection (DPI), mice from the PBS-treated group and the negative liposome control group developed obvious morbidity symptoms, such as convulsions and paralysis. All mice in these two groups died at 9 DPI (Figure 3B). Mice treated with 5 μg nCoV-L exhibited muscular paralysis of the rear legs at 5 DPI. One mouse died at 8 DPI. Mice treated with 10 or 15 μg nCoV-L developed mild symptoms, such as slow action at 5 DPI, which disappeared gradually within one week. No death occurred during the observation period. Therefore, treatment of 10 μg nCoV-L 1 h after viral infection prevented mouse death from lethal dosage infection of EV-71/C2-2.

Moreover, as shown in Figure 3A, the viral loads in different tissues of mice, including brains, livers, spleens, paraspinal muscles, hind leg muscles, and blood from each group, were evaluated at 5 DPI (Figure 3E). Viral loads in those tissues of mice from the 10 μg nCoV-L treated group were significantly lower (about 100-fold) than that in the control group (*p* < 0.05). EV71/C2-2 showed prominent tropism in muscle, and viral loads in the paraspinal muscles and hindquarters were significantly higher than that in the other tissues (*p* < 0.05). To directly assess the treatment effects of nCoV-L on immunopathology, histological analyses on HE-stained paraspinal and hind leg muscle sections from mice in each group were conducted. Muscular fibre necrosis and lymphocyte infiltration were observed in the paraspinal and hind leg muscles of mice treated with liposome. Mild muscular fibre necrosis, lymphocyte infiltration, and muscular fibres repair were observed in the nCoV-L-treated group (Figure A4D,E). The pathological experimental results inferred that nCoV-L had relatively good in vivo antiviral activity. Elisa assays were performed to analyse the IFN-β levels in mice treated with 10 μg nCoV-L for 3, 6, and 12 h. Compared with mice treated with liposome, the IFN-β level was significantly higher when treated with nCoV-L (*p* < 0.01) (Figure 3F). The highest IFN-β level was observed 3 h after treatment, then declined over time. In summary, nCoV-L effectively induced an IFN-I response and activated antiviral immunity in vivo.

### 3.4. Prophylactic Administration of nCoV-L Protects Mice from Lethal SARS-CoV-2 Virus Infection

The antiviral effect of nCoV-L against SARS-CoV-2 was also examined in this study. The lethal model constructed by McCray, et al. [28] was used. Briefly, K18-hACE2 mice were intranasally infected with 200 TCID_50_ of SARS-CoV-2/Delta. nCoV-L was injected intraperitoneally 6 h before or 0.5 h after infection (Figure 4A). The survival ratio and weight loss of the mice were shown in Figure 4B–D. Mice in the non-treated negative control group developed back arching and trembling symptoms at 4 DPI, and all mice died at 5 DPI (5/5). Only one mouse died when treated prophylactically with 30 μg nCoV-L (1/5) (Figure 4B), while the rest survived well. The prophylactic administration of nCoV-L not only protected mice from the challenge of a lethal dosage of SARS-CoV-2/Delta and relieved morbidity symptoms effectively but also inhibited weight loss in mice to some extent (Figure 4C). Administration of nCoV-L 0.5 h after virus infection protected mice from death (three out of five) but did not inhibit weight loss (Figure 4D). All surviving mice were killed by euthanasia at 5 DPI, when all mice in the control group died, to analyse the pathological state of lung tissue. Typical symptoms induced by SARS-CoV-2 infection, such as diffuse capillary congestion in the pulmonary vein and alveolar wall, necrosis and shedding of alveolar epithelial cells, and increased inflammatory cells in blood vessels, were found in lung tissues of mice without treatment [29,30]. Whereas prophylactic administration of 30 μg nCoV-L can protect mice from death, the surviving mice showed similar symptoms but with evidently less severe lesions (Figure 4E).

## 4. Discussion

Selection and reconstruction of sequences with hairpin structures from virus 5′ or 3′ UTRs are one of the primary research and development directions for developing RIG-I agonists [31]. A trefoil-type 5′PPP RNA fragment CV-B3 CL from the Human coxsackievirus B3 (CV-B3) virus inhibited the duplication of Dengue virus and Vesicular stomatitis Alagoas virus in vitro [32]. Chiang et al. [22] reported that M8, a fragment of 5′PPP RNA reconstructed from the Vesicular stomatitis Alagoas virus 5′UTR, had potent antiviral effects against multiple influenza viruses and Chikungunya virus in vivo and in vitro [33]. In this study, a fragment of 5′PPP RNA, nCoV-L, reconstructed from the SARS-CoV-2 5′UTR was designed and found to significantly inhibit infection of EV-71/C2-2, CV-B5/JS417, PV-1/Sabin and IAV/H9N2 in vitro (Figure 2 and Figure A3). In vivo, therapeutic administration of nCoV-L protected suckling mice from a lethal dosage of EV-71/C2-2 infection and mitigated morbidity symptoms (Figure 3). Prophylactic administration of nCoV-L also protected mice from a lethal dosage of SARS-CoV-2/Delta infection. The results demonstrate that nCoV-L has broad-spectrum antiviral effects. Recently, the role of activating the RLR/MAVS signalling pathways during DNA virus infection has been identified [34]. However, few studies have examined the effect of RIG-I agonists against DNA viruses, which will be investigated in further research.

This study examined the in vivo antiviral effects of nCoV-L against the SARS-CoV-2/Delta using the K18-hACE2 mouse lethal model. Prophylactic administration of nCoV-L generated a protective outcome to some extent. Pathologic changes were still observed in the lungs of the mice treated with nCoV-L; however, the symptoms were milder than those observed in the control group, indicating that the virus was not totally eliminated. This might be attributed to the immune escape of SARS-CoV-2/Delta, which encodes several proteins that inhibit the IFN-mediated signalling pathway [35,36]. The SARS-CoV-2 nucleocapsid protein (N) and membrane protein (M) were reported to play a role in suppressing host innate immunity by targeting the upstream event of RNA sensing [37,38]. SARS-CoV-2 non-structural protein 3 (NSP3) inhibits IFN-I production by cleaving the ubiquitin-like protein, ISG15, decreasing IRF3 phosphorylation, or directly cleaving IRF3 [39]. NSP1 can inhibit the translation of host mRNA, thus decreasing interferon and ISGs production [19]. Currently, the Omicron strain has overwhelmed the Delta strain to be the primary strain worldwide. Many known SARS-CoV-2 proteins inhibit the interferon response (NSP3, NSP6, NSP14, N, and M) have mutated in the Omicron strain. As a result, the Omicron strain provides weaker inhibition upon IFN signalling than Delta [39,40]. Thus, nCoV-L may provide a better antiviral effect against Omicron infection than Delta.

The experimental results also evaluated the effects of administration timing of nCoV-L on the antiviral outcomes in vivo. The results indicate that the in vivo protective effect of nCoV-L against EV-71/C2-2 virus is attenuated with delayed administration (Figure A4B,C). Consistent with the protective effect of SLR14, a similar result was observed when investigating the in vivo antiviral effect of nCoV-L against the SARS-CoV-2/Delta [41]. The therapeutic administration of nCoV-L could protect mice from the lethal dosage challenge of SARS-CoV-2/Delta and prolong survival time after viral infection. Prophylactic administration achieved a better protective effect, further reduced mouse mortality, and protected mice from weight loss after infection. It can be speculated that prophylactic administration of nCoV-L can induce a systematic interferon response to block the viral infection, thus improving the protective outcomes [18]. Hence, an appropriate immune strategy should be adopted to achieve a better protective effect.

This study proves that nCoV-L can activate the RIG-I receptor and induce interferon responses. Additionally, prophylactic administration of nCoV-L or administration in the early stage of infection protect mice from lethal viral infection. In conclusion, nCoV-L was found to generate a broad-spectrum antiviral effect by inducing an innate antiviral immune response, serving as a feasible antiviral strategy with great potential for future drug development prospects.

## Figures and Tables

**Figure 1 viruses-14-02451-f001:**
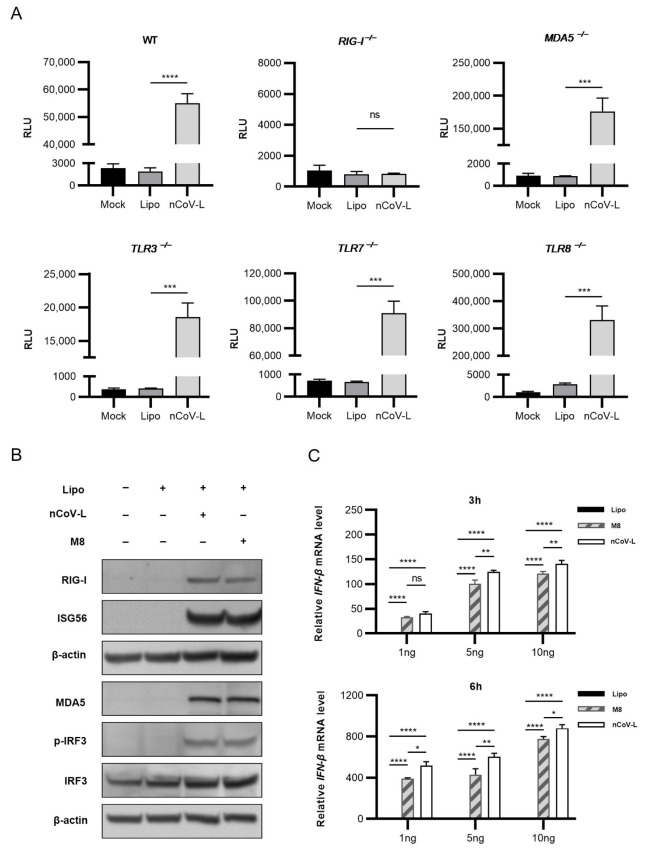
nCoV-L induces an interferon response by activating the RIG-I receptor. (**A**) 1 × 10^4^ wild-type (WT), *RIG-I ^−/−^, MDA5 ^−/−^, TLR3 ^−/−^, TLR7 ^−/−^, TLR8 ^−/−^*, and HEK293 cells expressing *ISG54*-luciferase reporter gene were cultivated in 96-well plate for 12 h, and then transfected with 50 ng nCoV-L using Lipofectamine 3000 (short for Lipo) or treated with equal dosage of Lipo as the negative control. Relative luciferase activity was tested after 24 h. N = 3, RLU indicates relative light unit, bar indicates SD. (**B**) 2 × 10^6^ A549 cells were transfected with 1 ng/mL nCoV-L or M8 using Lipo. Cell lysis was prepared and subjected to Western blot analysis 24 h after transfection. (**C**) 2 × 10^6^ A549 cells were transfected with 1 ng/mL,5 ng/mL or 10 ng/mL nCoV-L or M8 using Lipo. Cells were harvested 3 h or 6 h after transfection and total RNA was extracted. Relative *IFN-β* gene expression levels were determined by RT-qPCR. N = 3, bar indicates SD. ns, *p* > 0.05, *, *p* < 0.05, **, *p* < 0.01, ***, *p* < 0.005, ****, *p* < 0.001.

**Figure 2 viruses-14-02451-f002:**
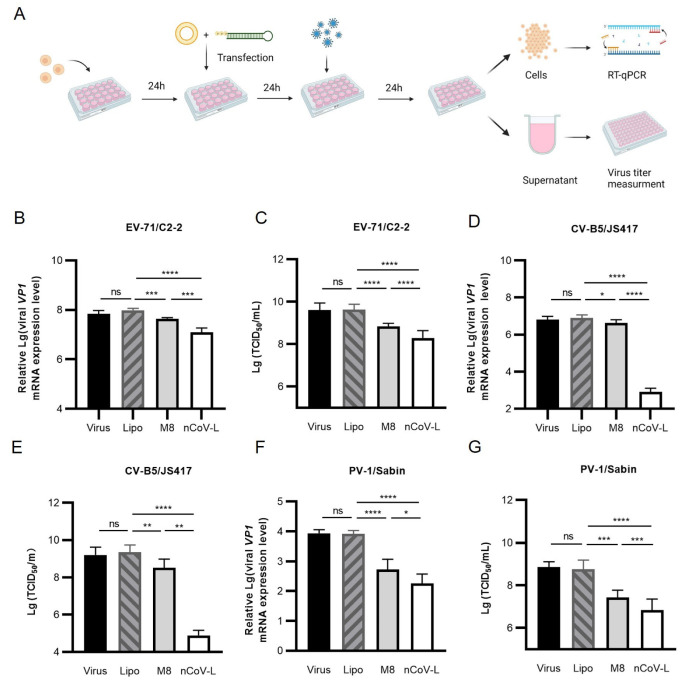
The in vitro antiviral effect of nCoV-L against three different RNA viruses. (**A**) Experimental scheme. 2 × 10^5^ cells were transfected with nCoV-L or M8 for 24 h and then infected with virus (0.1 MOI) for 24 h. Cells were harvested, and total RNA was extracted. Viral loads were measured by RT-qPCR. Viral titers in cell culture supernatants were also determined. RD cells were transfected with 30 ng/mL nCoV-L or M8, and then infected with EV-71/C2-2. Viral loads (**B**) and viral titers (**C**) are shown. c2c12-CAR cells were transfected with 10 ng/mL nCoV-L or M8, and then infected with CV-B5/JS417. Viral loads (**D**) and viral titers (**E**) are shown. HEp-2 cells were transfected with 10 ng/mL nCoV-L or M8, and then infected with PV-1/Sabin. Viral loads (**F**) and Viral titers (**G**) are shown. N = 3, bar indicates SD. ns, *p* > 0.05, *, *p* < 0.05, **, *p* < 0.01, ***, *p* < 0.005, ****, *p* < 0.001.

**Figure 3 viruses-14-02451-f003:**
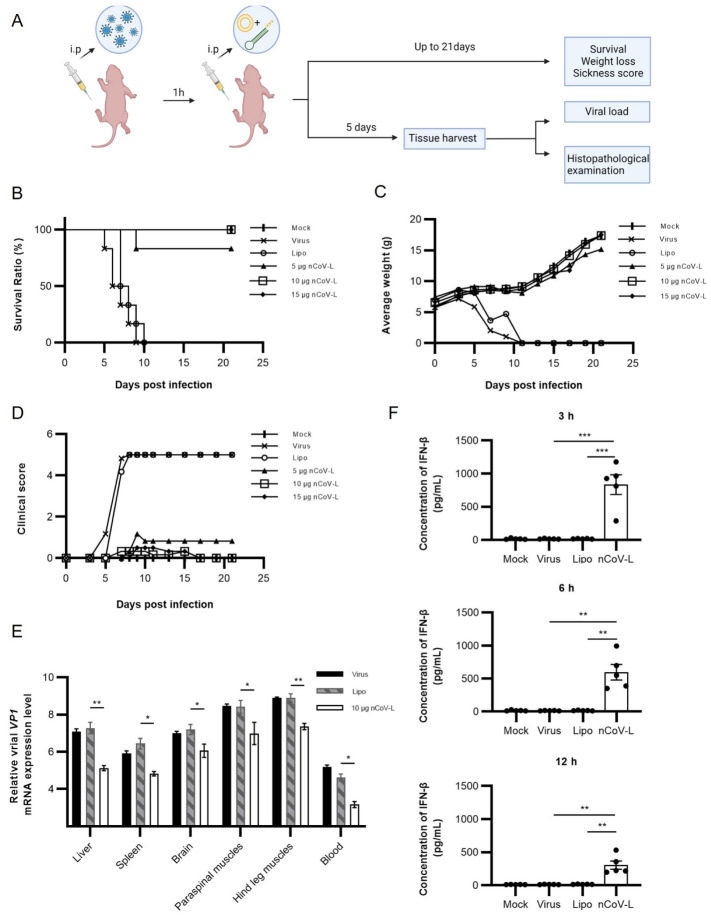
The in vivo antiviral effect of nCoV-L against EV71/C2-2. (**A**) Experimental scheme. 7-day-old BALB/c suckling mice (N = 6) were challenged intraperitoneally with 1.26 × 10^6^ TCID_50_ EV-71/C2-2. Mice were injected intraperitoneally with PBS, Lipo, 5 μg, 10 μg, or 15 μg nCoV-L complexed with Lipo 1 h later. Survival (**B**), average weight (**C**), and sickness (**D**) were monitored or scored every two days up to 21 DPI. In a separate cohort (N = 3, 10 μg nCoV-L), blood, liver, spleen, brain tissues, paraspinal, and hind leg muscle were collected for virological and immunological analysis at 5 DPI. (**E**) Viral loads in the above tissues were measured by RT-qPCR. Suckling mice were injected intraperitoneally with PBS, 1.26 × 10^6^ TCID_50_ EV-71/C2-2, Lipo or 10 μg nCoV-L complexed with Lipo. Blood was collected 3, 6, 12 h after injection (N = 5). (**F**) Concentration of IFN-β in serum were measured by ELISA. N = 5, Bar indicates SD. *, *p* < 0.05, **, *p* < 0.01, ***, *p* < 0.005.

**Figure 4 viruses-14-02451-f004:**
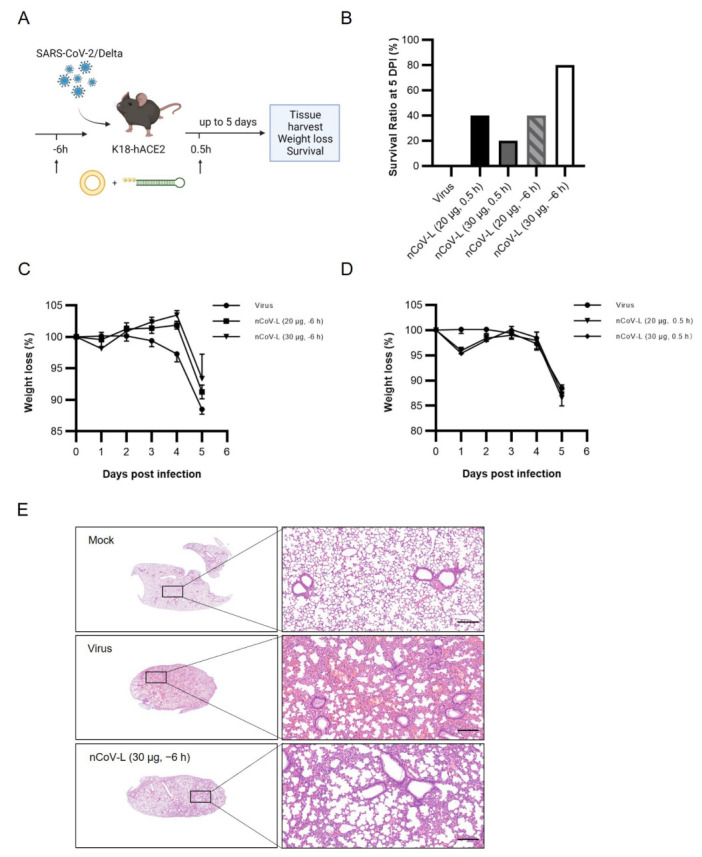
The in vivo antiviral effect of nCoV-L against SARS-CoV-2. (**A**) Experimental scheme. K18-hACE2 mice were intranasally infected with 200 TCID_50_ SARS-CoV-2/Delta. 20 μg or 30 μg nCoV-L complexed with Lipo were intraperitoneally injected 6 h before or 0.5 h after infection. Survival and weight loss were monitored daily up to 5 DPI when mice infected with virus were all dead. (**B**) Survival ratio of each group at 5 DPI are shown. (**C**) Weight loss of mice treated with nCoV-L or Lipo 6 h before infection. (**D**) Weight loss of mice treated with nCoV-L or Lipo 0.5 h after infection. Bar indicates SEM. (**E**) HE staining of lung from mock, PBS treated or 30 μg nCoV-L prophylactically treated mice at 5 DPI are shown. Scale bar represents 200 μm.

**Table 1 viruses-14-02451-t001:** sgRNA sequences.

Gene	sgRNA
*TLR3*	CAACTTTCTTGGGACTAAAG
*TLR7*	TTCAGCATGTGCCCCCAAGA
*TLR8*	TTAGTGGGAGAAATAGCCTC
*MDA5*	GCTCAGGCCTTACCAAATGG
*RIG-I*	AATTCCCACAAGGACAAAAG

**Table 2 viruses-14-02451-t002:** Primer sequences used for RT-qPCR.

Gene	Direction	Primer Sequence (5′-3′)
*EV71/C2-2*	Forward	TAG TTT CTT CAG CAG GGC GG
	Reverse	ACC ATT GGT AAG CAC TCG CA
*CV-B5/JS417*	Forward	GGT GTC CGT GTT TCC TTT TAT TCC TAC
	Reverse	CAA GTA GAT AAT AGC TCT GTT TGT CAC CG
*PV-1/Sabin*	Forward	TGA TCA CAA CCC GAC CAA GG
	Reverse	TAA TCC ACT CCA GGG CCG TA
*IAV/H9N2*	Forward	CCG GAA TTT CTG GAG AGG CG
	Reverse	TAC ACA AGC AGG CAA GCA GG
*IFN-β*	Forward	GCT TGG ATT CCT ACA AAG AAG CA
	Reverse	ATA GAT GGT CAA TGC GGC GTC

## Data Availability

All data, generated or analysed, and materials during this study are included in this published article.

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
