# Peer review of "A Short 5′triphosphate RNA nCoV-L Induces a Broad-Spectrum Antiviral Response by Activating RIG-I"

_viruses, 2022, doi:10.3390/v14112451_

Round 1
Reviewer 1 Report
This manuscript describes a short 5’-triphosphate RNA, derived from a stem-loop structure in the 5’UTR of SARS-CoV-2, and its use as an activator of RIG-I, to induce an interferon response and inhibit the replication of several enteroviruses and SARS-CoV-2, both in cell culture and in vivo. The RNA, named by the authors as nCoV-L, protected suckling mice from challenge with enterovirus 71 and partially protected transgenic K18-hACE2 mice from SARS-CoV-2.
The experiments are done well and I think the results deserve to be published. I do, however, have some reservations regarding the novelty of this work. Other than the sequence, how does the nCoV-L RNA differ from M8 and SLR14 described in the authors’ citations. Any difference in length, stability, structure? Why would one use nCoV-L, rather than M8, SLR14, or other similar RNAs? Please justify.
The text does contain some typos and grammatical errors, which are understandable. I’ve pointed out some of these errors under minor points, but I suggest the text could do with a further proof-reading.
Major points:
Line 81: How were the CRISPR knockout lines characterized to confirm knockout? Please describe.
The authors test the activity of nCoV-L against 3 different enteroviruses, as well as SARS-CoV-2. Any additional, even in vitro, testing against viruses of other families would help the authors’ characterization of the molecule as being “broad-spectrum” – I question whether 2 families (both positive-strand) constitute a spectrum.
Minor points:
Line 80: replace “The” with “the”.
Line 82: “designed and” do not need italicized letters.
Line 83: replace “cas9” with “Cas9”.
Line 325: replace “duplication” with “infection”.
Line 349: “…the Omicron strain provides weaker inhibition upon IFN signalling than delta.” Please provide a citation for this observation.
Line 355: Suggest rewording this sentence. As written, this could be interpreted to mean that Mao et al already tested nCoV-L, whereas they actually tested a different RNA molecule.
Fig. 4C: Are the weights statistically significantly different?
Fig. A1A: This isn’t a very helpful diagram! How long is the stem and how long is the loop? Maybe these could be marked on to make it a bit more useful. Is this the result from the RNAfold analysis mentioned in line 126? Maybe mention this in the legend.
Fig. A1 Legend: replace “fitures” with “features”.
Reviewer 2 Report
In this study, the authors constructed a short pppRNA (named nCoV-L) according to the sequence of SARS-CoV-2 5’-UTR and suggested that nCoV-L could activate RIG-I signalling specifically and protect cells and animals from viral infections. nCoV-L was claimed to have a broad-spectrum antiviral potential. However, some conclusions are not supported by strong experimental evidence. Appropriate controls are missing in some experiments. The manuscript could be significantly improved if the following specific points are addressed satisfactorily.
1. For Fig 1, poly(I:C) should be included as positive control for comparison.
2. Line 195 claimed “nCoV-L can specifically activate the RIG-I receptor” but from Fig 1B, it seems that MDA5 was also activated by nCoV-L as seen in the blot. Other than WT cells, RIG-I-KO and MDA5-KO cells should also be tested to verify this.
3. Fig 1C, the dosage effect should be investigated for better comparison of the potency of activation.
4. Fig 1, why is the amount of nCoV-L transfected in panels 1A, 1B and 1C so different (50ng vs 1ng). This makes it difficult to correlate results from different panels.
5. Line 262-268, are the authors referring to the results in Fig A2 D and E?
6. For the analysis of in vivo effect of nCoV-L against SARS-CoV-2 in Fig 4, it is not clear why the mice have to be sacrificed by Day 5. Monitoring of treated mice beyond 5 days post-infection is necessary to determine whether nCoV-L has significant protective effect against SARS-CoV-2 infection. Specifically, fig 4B should be replaced by a survival curve over a longer duration. Fig 4C and 4D, mice should be monitored beyond 5 dpi. Viral loads in various tissues at different time points are also essential.
7. Line 347-350, please provide references that clearly suggested the weakening of inhibition against IFN signalling in Omicron strain due to specific mutation of SARS-CoV-2 proteins.
8. Ling 394, what is “Fitures”?
9. Fig A2D, does the labelling of “virus” mean “PBS-treated control”? It is confusing. Furthermore, mock treated samples (No infection control) should be analysed for a meaningful comparison.
Round 2
Reviewer 1 Report
The authors have addressed my concerns.
Author Response
Thanks for your affirmation!
Reviewer 2 Report
The authors have satisfactorily addressed the concerns that I have raised previously.
However, I do feel that the authors could actually incorporate some additional results in the reply letter into the manuscript as far as possible. If not replacing the original figures, they may still put them into the supplementary and further elaborate in the main text. I believe this will be of great benefit to the readers.
Author Response
Thanks for your suggestion, we have incorporated some additional results into the manuscript, hoping the revision is much improved. All the changes were marked in red.
